# Fluorescence intermittency originates from reclustering in two-dimensional organic semiconductors

Anthony Ruth[1], Michitoshi Hayashi[2], Peter Zapol[3], Jixin Si[1], Matthew P. McDonald[4], Yurii V. Morozov[5], Masaru Kuno[5] & Boldizsár Jankó[1]

Fluorescence intermittency or blinking is observed in nearly all nanoscale fluorophores. It is characterized by universal power-law distributions in on- and off-times as well as $1/f$ behaviour in corresponding emission power spectral densities. Blinking, previously seen in confined zero- and one-dimensional systems has recently been documented in two-dimensional reduced graphene oxide. Here we show that unexpected blinking during graphene oxide-to-reduced graphene oxide photoreduction is attributed, in large part, to the redistribution of carbon $sp^2$ domains. This reclustering generates fluctuations in the number/size of emissive graphenic nanoclusters wherein multiscale modelling captures essential experimental aspects of reduced graphene oxide's absorption/emission trajectories, while simultaneously connecting them to the underlying photochemistry responsible for graphene oxide's reduction. These simulations thus establish causality between currently unexplained, long timescale emission intermittency in a quantum mechanical fluorophore and identifiable chemical reactions that ultimately lead to switching between on and off states.

[1] Department of Physics, University of Notre Dame, Notre Dame, Indiana 46556, USA. [2] National Taiwan University, Center for Condensed Matter Sciences, National Taiwan University,, Taipei 10617, Taiwan. [3] Materials Science Division, Argonne National Laboratory, Argonne, Illinois 60439, USA. [4] Max Planck Institute for the Science of Light, Erlangen, Germany. [5] Department of Chemistry and Biochemistry, University of Notre Dame, Notre Dame, Indiana 46556, USA. Correspondence and requests for materials should be addressed to B.J. (email: bjanko@nd.edu).

Observing chemical reactions as they occur reveals unforeseen connections. An *in situ* fluorescence experiment monitoring single cholesterol oxidase molecules observed non-Markovian rates of reversible reactions. This indicates a form of memory in the system which influences future reaction dynamics[1].

For decades, researchers have been trying to determine the finite states of a system undergoing fluorescence intermittency, a phenomenon in which a fluorophore consistently cycles between power-law distributed bright periods and power-law distributed dark periods. Many models have been developed to explain the contrast between fast radiative recombination (ns) and slow timescales seen in fluorescence intermittency (s–min)[2–6]. One of the first and most widely used microscopic models assumes that the dark state is charged[7,8], and it is often invoked to justify the existence of long intermittent periods in colloidal quantum dot blinking trajectories. In this model, an on-to-off transition occurs due to Auger ionization of a nanocrystal. The implicit assumption is that a charged quantum dot remains non-emissive until neutralized by return of the ejected electron. Although some of the predictions of the original Efros–Rosen model are in contradiction with experiments[9,10], recent work by Osad'ko *et al.*[8] showed that a modified version of the charging model is consistent with power-law kinetics. However, as shown by Guyot-Sionnest[9], charged quantum dots are emissive, and alternative blinking scenarios also have to be considered. Quantum dot blinking has also been attributed to light-induced surface ligand reorganization[11] and even to light-induced defect rearrangement in nanocrystals[12]. Beyond quantum dots, blinking has been observed in a host of other systems. Examples includes instances of fluorescence intermittency in single molecules[13] and in single polymer strands[14]. Explanations for blinking in these other systems hence range from surface electron (hole) traps to oxidation. Consequently, within the broader context of single-particle microscopy, and even within the realm of quantum dot blinking, there exists no consensus as to the microscopic origin of blinking.

Graphene oxide (GO) is a 2D material formed by the oxidation of graphene[15]. Unlike graphene, GO possesses a band-gap that gives rise to strong fluorescence in the visible spectrum[16] that can be tuned by gradual reduction into reduced graphene oxide (rGO)[17]. Prior single sheet absorption and emission microscopy/spectroscopy studies[18,19] have shown that this novel system's photophysical properties evolve dramatically during photoreduction; GO's emission first quenches and then brightens under continuous 405 nm irradiation. This behaviour is linked to reduction monitored through synchronous absorption microscopy, emission spectroscopy, and ensemble characterization techniques[18,19]. Intriguingly, the brightening phase is punctuated by spatially heterogeneous fluctuations in emission intensity with $1/f$ power spectral densities (PSDs) typical of fluorescence intermittency (cf. Supplementary Movie 1)[20]. Although the multiple recombination center model[21] provides a phenomenological framework for the $1/f$ power spectrum observed experimentally, it does not provide a microscopic mechanism for blinking. Our paper provides a definitive link between blinking in rGO and photostimulated carbon etching.

In the current study, concerted single-particle optical microscopies are coupled to multiscale spatial (nm²–mm²) and temporal (ms–h) simulations of relevant photolytic reactions to demonstrate how the chemical reactivity of individual graphene oxide sites leads to different reduced graphene oxide domain structures exhibiting distributed absorption and fluorescence properties. The agreement between simulation and experiment additionally grows with model parameterization until new optical behaviour, fluorescence intermittency, emerges from rules governing single chemical reactions. This has broader implications beyond GO/rGO blinking since despite the nearly three decades of work on the matter[22,23], the current study is the first to link blinking to a definitive chemical process. To determine if rGO's blinking and reduction mechanism are implicitly connected, we have carried out multiscale theoretical modelling of GO-to-rGO interconversion. Figure 1 shows density functional theory (DFT)/time-dependent density functional theory (TDDFT) and Kinetic Monte Carlo (MC) simulations which connect the photoreduction-induced, structural evolution of GO to the optical response of its individual $sp^2$ domains. GO features a $sp^2$-coordinated carbon honeycomb lattice, which contains vacancies, defects and functional groups distributed across its basal plane and edges. Different models[15,24–27] suggest that epoxide (COC) and hydroxide (COH) groups predominantly decorate GO's basal plane while edges are carbonyl- (CO), carboxyl- (COOH) or hydrogen-terminated. Consequently, graphenic domains are embedded within a disordered, $sp^3$-hybridized matrix with photolytic reduction leading to GO-to-rGO interconversion through changes to the local number, size and overall density of aromatic $sp^2$ clusters[15,17,28].

## Results

**Optical properties of individual graphenic domains.** DFT/TDDFT[29] are first used to establish minimum transition energies, absorption cross-sections/emission intensities and quantum yields (QYs) for a series of successively larger graphenic clusters: $C_{24}H_{12}$, $C_{54}H_{18}$, $C_{96}H_{24}$, $C_{150}H_{30}$ and $C_{216}H_{36}$. Clar's pioneering work[30] demonstrated that the optical response of polyaromatic hydrocarbons (PAHs) depends exquisitely on the number of carbon atoms as well as the maximum number of possible aromatic sextets. Different PAHs, which conserve the number of carbon atoms and aromatic sextets, possess near identical optical properties. Consequently, the chosen graphenic clusters enable construction of a calibration curve, linking $sp^2$ domain size to QY and spanning the visible spectrum. Corresponding graphenic cluster emission energies (QYs) are 4.16 eV (31.8%), 2.90 eV (12.5%), 2.18 eV (4.52%), 1.69 eV (1.25%), and 1.32 eV (0.092%) respectively[16,31,32].

**Simulation of photoreduction with kinetic Monte Carlo.** MC simulations then model the evolution of GO's graphenic clusters during photoreduction (cf. Supplementary Figs 2 and 3). A 20 nm × 20 nm carbon honeycomb lattice is first generated whereupon functional groups are incorporated into the lattice by randomly placing hydroxide and epoxide groups across the basal plane. Three chemical reactions are simulated to evolve the lattice during reduction: hydroxyl abstraction equation (1), 1,3 epoxide abstraction equation (2) and direct carbon sublimation equation (3) (refs 33,34).

$$C_nOH \rightarrow C_n + OH, \text{ (Rate Constant} = k_{OH}) \quad (1)$$

$$C_nO \rightarrow C_{n-1} + CO, \text{ (Rate Constant} = k_O) \quad (2)$$

$$C_n \rightarrow C_{n-1} + C, \text{ (Rate Constant} = k_C) \quad (3)$$

Although the actual chemical reactions which occur during GO photoreduction are more complicated and numerous[34], equations (1–3) produce photoreduction and lattice disintegration which dictate GO's long-term structural evolution[15,16,28]. Additional details regarding the parameterization can be found in Supplementary Notes 1 and 2.

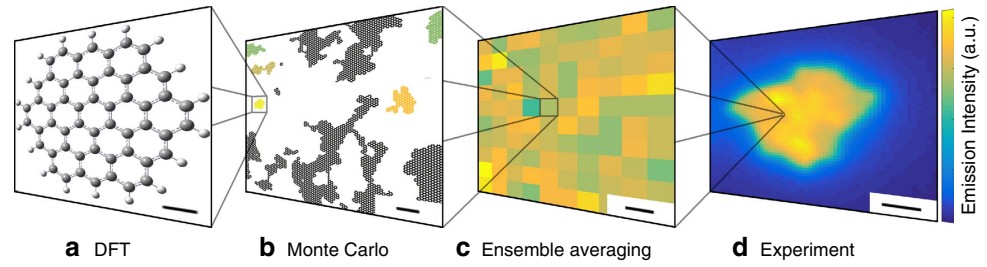

**Figure 1 | Cascade of domain and length scales used in this study.** Optical properties of each size range are connected to the next larger size range. Colour is used to represent emission intensity as indicated by the scale bar on the right. (**a**) Typical domain used in DFT calculations to characterize the absorption/emission properties of individual carbon nanoclusters; (**b**) Structure of GO at a specific time-step generated by MC simulation; (**c**) A 200 × 200 nm composite of 100 MC-simulated domains equivalent to the optical response of one pixel monitored in the experiment; (**d**) time-averaged emission intensity for one sample used in the study. Scale bar, 0.2 nm (**a**); 2 nm (**b**); 20 nm (**c**); and 2 μm (**d**).

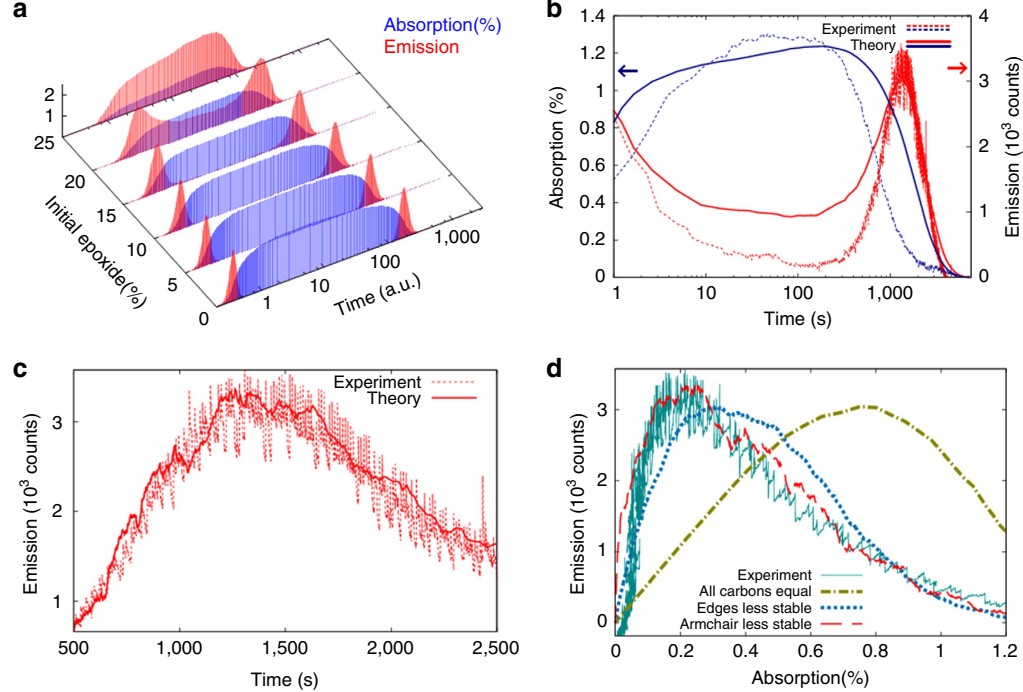

**Figure 2 | Comparison of experimental and MC time trajectories.** (**a**) Theoretical emission/absorption trajectories for different epoxide concentrations determined by the ratio of epoxide to total carbon content (COC:C). (**b**) Typical experimental emission/absorption trajectory compared to the best theoretical trajectory from **a** (ref. 18). (**c**) Close-up of emission profile during the blinking phase. (**d**) Emission versus absorption correlation plots.

**Optical properties of clusters in MC.** DFT/TDDFT results are incorporated into MC simulations. Specifically, graphenic clusters, possessing suitable emission energies and QYs to be experimentally observed, are monitored. The absorption of an individual graphenic domain is proportional to its $sp^2$ carbon content while its corresponding emission intensity is the product of its absorption cross-section and associated QY. $C_{24}H_{12}$ and $C_{216}H_{36}$ have emission energies outside the visible range. Therefore, only clusters in the range of 50 to 150 carbon atoms contribute significantly to the observed emission (cf. Supplementary Table 1; Supplementary Fig. 1). Counter intuitively, smaller clusters emit more despite their lower absorption cross-section. By adding the response of all spectro-scopically relevant graphenic clusters within suitably large areas of the GO lattice, theoretical emission/absorption intensity time trajectories are obtained, which can be directly compared to experiment[18].

**Emission and absorption time trajectories.** Figure 2a shows emission/absorption trajectories resulting from using equations (1–3) ($k_{OH} > k_O > k_C$) in the MC simulation. The percentage of epoxide:carbon (COC:C) has been varied from 0 to 25% to find the trajectory in best agreement with experiment. This yields an optimal 20%, consistent with previous studies showing C:O ratios between 4:1 and 2:1 if one assumes that half of the oxygen is in epoxide form ref. 15. In all cases, remaining $sp^2$ sites have been OH decorated.

Figure 2b compares an experimental emission/absorption trajectory to the best match from Fig. 2a (20% epoxide). This value was chosen by comparing absorption maximum and valley-to-peak emission intensity ratio. Evident in either case is that the emission exhibits an initial decay, followed by a low plateau. Meanwhile, the absorption exhibits an initial rise followed by a high plateau. The emission then exhibits a second photobrightening peak with blinking prior to apparent

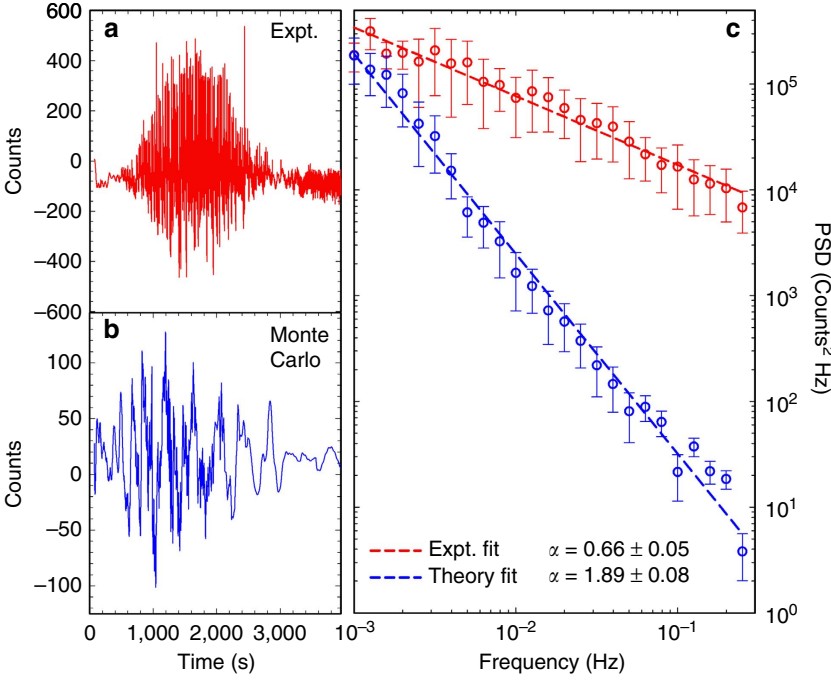

**Figure 3 | Frequency domain analysis of emission trajectories.** (**a**) Fourier filtered experimental trajectory. (**b**) Fourier filtered MC trajectory. In both cases, frequencies below 0.01 Hz have been removed. The envelope of the experimental Fourier filtered trajectory grows around ca. 500 s and decays around ca. 3,000 s. The magnitude of the Monte Carlo Fourier filtered trajectory grows and decays at nearly the same time as the experiment. (**c**) Power spectral density of experimental and MC trajectories. Error bars are s.e.m. The error in the power-law exponent reflects the quality of the fit and shows that both experimental and MC trajectories fit well to power-law distributions.

photobleaching and irreversible sample damage. While the emission is intermittent, the absorption follows a smooth decay[18].

The calculated emission trajectory remarkably reproduces the two-peaked structure of the experiment. In addition, the optimal epoxide percentage simultaneously reproduces the maximum absorption value of the experiment (1.2%) and the emission valley-to-peak ratio. The simulations indicate that the first peak in the emission trajectory arises from OH removal, equation (1). The plateau stems from COC removal, equation (2). The final photobrightening peak occurs due to the sublimation of carbon, equation (3) (cf. Supplementary Fig. 4; Supplementary Movie 2).

**Local structure model of carbon sublimation**. Given that blinking occurs during carbon sublimation, we focus on more realistically modelling this phenomenon by allowing carbon sublimation rate constants to differ. This accounts for the varied local structure found in GO/rGO (for example, singly coordinated sites versus armchair edges, zigzag edges and triply coordinated sites). Specifically, we set $k_{C1} > k_{Carm} > k_{Czig} > k_{C3}$ in equation (3), and this parameterization is used in simulations yielding Figs 2c,d; 3 and 4.

Figure 2c shows the resulting superb agreement with experiment. Not only is clear intermittency evident in the calculated trajectories but the magnitude of intensity fluctuations is comparable to experiment.

Since emission and absorption are intimately tied to $sp^2$ domain structure, the improved emission versus absorption correlation shown in Fig. 2d, reveals that the local structure model for carbon sublimation reproduces the number and size distribution of $sp^2$ domains that were created in the experiment both leading up to and during blinking.

**Blinking from carbon sublimation**. In Fig. 3, the frequency domain has been used to elicit features of fluorescence intermittency in the experimental and MC trajectories. Figure 3a,b show the trajectories after Fourier filtering. This process removes the long-term evolution of GO's emission and more clearly reveals the fluorescence intermittency that is present. Blinking begins and ends at approximately the same time in the MC and the experiment. This leads us to conclude that the MC simulation captures the key difference between the stable emission and the fluorescence intermittency phase of the system.

Subsequent analyses of their power spectral densities reveal that both simulated and experimental trajectories exhibit $1/f^\alpha$ power-law behaviours (Fig. 3c). Experimental and theoretical power-law exponents of $0.5 < \alpha < 0.9$ and $\alpha = 1.9$ are found, which fall within the range of exponents (that is, 0.5 to 2.0) typically seen for other nanoscale emitters exhibiting fluorescence intermittency[2]. The quantitative discrepancy between experiment and theory is narrowed by including Poisson noise. In this regard, Poisson noise accounts for counting statistics and also provides a simple model for intensity fluctuations arising from reversible reactions. The latter, in particular, stems from dynamic changes to the number of emissive domains, which contribute to the overall observed emission intensity. Including counting statistics to the Monte Carlo simulation reduces its power-law exponent to 0.7 , which is closer to the range of exponents seen in the experiment. Furthermore, after including cluster fluctuations to model reversible processes, we recover substantial spectral weight at high frequencies. Additional details regarding the effect of Poisson noise on the Monte Carlo results and the methods used to calculate PSDs can be found in Supplementary Figs 5 and 6; Supplementary Note 3; Supplementary Method 3.

In whole, the model reproduces the two peaked absorption/emission trajectories in Fig. 2b. Subsequently accounting for local structure immediately reproduces the emission versus absorption

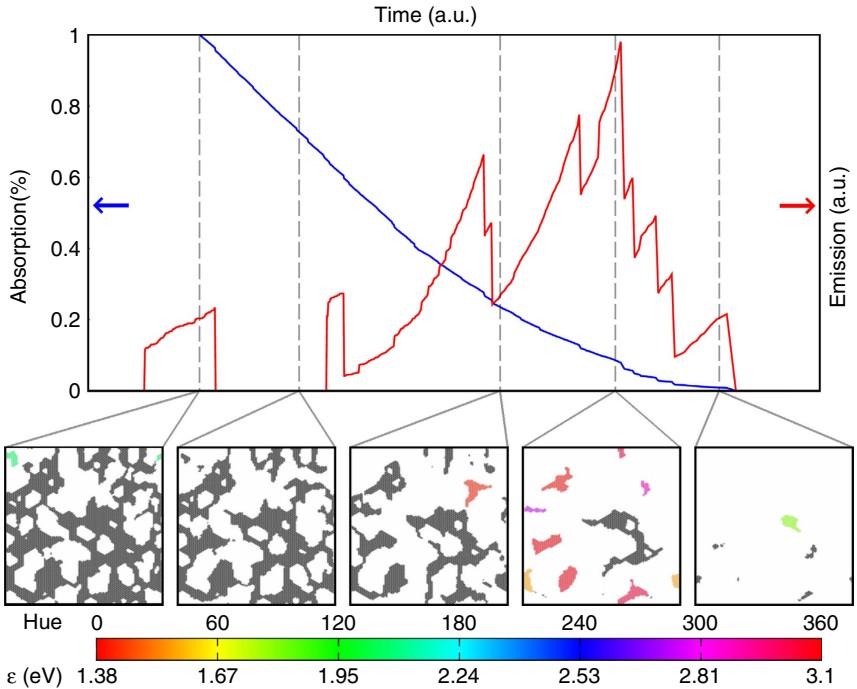

**Figure 4 | Simulated emission, absorption and domain structure during rGO blinking.** Cluster colour coding reflects the energy of light emitted by clusters. Black $sp^2$ domains do not emit although they do absorb light. Supplementary Movie 3 illustrates the structural evolution during blinking.

correlation seen experimentally in Fig. 2d and the appearance of emission intermittency seen in Figs 2c,d and 3. This last success is especially important, because it points to a structural mechanism, which underlies blinking in graphene oxide.

**Reclustering mechanism of blinking**. To explore deeper mechanistic aspects of GO/rGO blinking, Fig. 4 highlights specific features of emission/absorption trajectories and associated structure in a 20 nm × 20 nm area. MC simulations illustrate that relatively few spectrally relevant clusters exist per unit area in GO/rGO during the blinking segment of its trajectories. In fact, approximately one emissive cluster exists per 100 nm² at any given time. Reclustering of these graphenic domains is therefore responsible for corresponding fluctuations in the associated emission intensity (for example, Fig. 4). The structure at subsequent time steps reveals how emissive domains separate from large networks and subsequently photobleach. This explains why blinking shows up in emission but not absorption. Blinking is thus a consequence of an evolving equilibrium between the photolytic separation and disintegration of graphenic clusters of the correct, spectrally relevant size. Such light-induced structural changes are also the likely cause of blinking in other fluorophores[35] although there may not be a universal mechanism for blinking.

**Discussion**

In summary, a multiscale numerical simulation of GO–rGO photoreduction reproduces essential experimental features characterizing the evolution of GO's optical properties. Specifically, the modelling successfully explains experimentally observed emission/absorption trajectories with strong temporal fluctuations about a second photobrightening feature. In addition, both experimental and simulated PSDs exhibit power-law behaviour characteristic of universal emission intermittency seen other fluorophores. The explicit link established between emission/absorption trajectories and the structural/chemical

transformation of GO/rGO strongly suggests that blinking in rGO originates from reclustering—$sp^2$ cluster creation and destruction processes as well as processes which distort $sp^2$ domains in a reversible manner. Reclustering alters the size, shape and ultimately the QY of underlying graphenic nanoclusters leading to emergent emission intermittency. This conclusion is especially important from a mechanistic standpoint since it has been a longstanding mystery[12,36] as to how an inherently quantum mechanical system—such as a quantum dot, rod, wire, but also rGO –exhibits fluctuations over timescales much longer than those of fundamental electronic processes.

**Methods**

In order to provide molecular insight into optically measured experimental data, the developed model scales six orders of spatial magnitude in a series of steps. First, DFT and TDDFT is carried out to determine gap energies, radiative recombination rates, nonradiative recombination rates, and thus QYs on Å–nm scaled fluorescent domains (cf. Supplementary Fig. 1). The domains we studied via DFT/TDDFT span the entire visible range and are representative of all $sp^2$ domains which contribute to the photoluminescence of graphene oxide. Next, MC simulations use this information to model larger regions of a GO sheet during dynamical chemical reactions. During the simulation, cluster detection is used to count the number of connected carbon atoms; the gap energy, associated absorption cross-section and QY expressions are applied to calculate the colour, absorption and emission intensity of that cluster. Clusters whose gap energy exceeds the photon excitation energy are excluded. The absorption cross-section and emission intensity of all clusters within the region is summed up. The absorption is rescaled to calculate a per cent absorption. Finally, in order to simulate a region comparable to a pixel in the experiment many small regions are simulated and the results are averaged.

All DFT calculations were performed with the Gaussian 09 software package. Structures were relaxed to a force of less than 0.01 eV Å⁻¹ before optical calculations were performed. All calculations were performed with the 6–31G* basis set at the B3LYP (refs 37–40) level. QYs are calculated by comparing the rate of radiative recombination with the total recovery rate to the ground state (cf. Supplementary Methods 1 and 2). The absorption in MC simulations was calculated by comparing the number of $sp^2$ carbon atoms in the simulation with a piece of graphene of the same area. The ratio was then multiplied by 2.3% (The absorption of graphene at 520 nm)[32]. Emission QYs of the five selected clusters were calculated by starting with the lowest energy level, which has a significant dipole element to the ground state and then comparing photoluminescence rates to internal conversion rates. The gap energy, absorption cross-section and emission QY of individual clusters formed in MC simulations

was subsequently estimated through TDDFT-established calibration curves, $\varepsilon(eV) = \frac{20.5}{\sqrt{N}}$, $\sigma\,(a.u.) = N$, and $QY(\%) = 274.5e^{-0.44\sqrt{N}}$, where $N$ is the number of carbon atoms in the cluster. Kinetic Monte Carlo simulations were performed with 17,280 carbon lattice sites (21.3 nm × 21.3 nm) with larger regions represented by averaging many such simulations. Each kinetic step removed a carbon atom/functional group from the simulation, and the reaction to be performed was chosen randomly with probability proportional to rate, rate constant times number of reactants, and site of reaction chosen randomly.

**Synthesis of graphene oxide.** GO was synthesized using a modified Hummers method[19]. In brief, 300 mg of graphite, 36 ml of $H_2SO_4$ and 4 ml of $H_3PO_4$ were combined in an ice bath and continuously stirred for 4 h. Next, 3.6 mg of $KMnO_4$ was added and the mixture was stirred for an additional 48 h. The temperature was kept below, 25 °C for this period. The suspension was then diluted to double the volume with deionized water and subjected to sonotrode sonication at 20 kHz for 1.5 h. Finally, 3% $H_2O_2$ was added to the suspension until it turned yellow, indicating that highly oxidized graphite was produced. Upon centrifugation, the precipitate was washed with 1 M HCl and then repeatedly washed with deionized water. The resulting red-brown suspension was then freeze-dried for storage, yielding a cream coloured solid[19].

**Individual GO sheet microscopy.** Dilute GO/ethanol suspensions were sonicated at 42 kHz for ∼10 s and subsequently drop-cast onto flamed fused silica microscope coverslips. This produced a sample coverage of ∼1 single layer GO sheet per several $\mu m^2$, with flake sizes ranging from 1 to 10 μm. Individual sheets were imaged with a home-built inverted microscope utilizing a continuous wave 405 nm laser. The excitation source was introduced to the sample in an epi-illumination arrangement, with an associated excitation intensity of 380 W cm$^{-2}$ and an ∼30 μm excitation spot. Fluorescence was captured with an electron-multiplied charge-coupled device (EM-CCD) via a collection objective (Zeiss, 1.4 NA) and imaging lens ($f = 160$ mm) for a final resolution of ∼200 nm per pixel[19].

**Data availability.** The Monte Carlo simulation code is available at: https://github.com/aruth2/GOMonteCarlo. The data that support the findings of this study are available from the corresponding author upon request.

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

## Acknowledgements

A.R. acknowledges support from a NASA Space Technology Research Fellowship. M.K. thanks the Army Research Office (Grant No. W911NF-12-1-0578) for financial support. The work of P.Z. was supported by U.S. Department of Energy, Office of Science, Office of Basic Energy Sciences, Division of Materials Science and Engineering under Contract No. DE-AC02-06CH11357. M.H. thanks Ministry of Science and Technology, Taiwan (Dragon Gate Program 103-2911-I-002-595) for financial support. This work was also supported by the Department of Physics, the Department of Chemistry and Biochemistry, and the College of Science of the University of Notre Dame.

## Author contributions

Experimental data collected by M.P.M., Y.V.M. and M.K. DFT performed by M.H. Model created by P.Z., A.R., M.K. and B.J. Monte Carlo simulations coded by A.R. PSD analysis from J.S. Manuscript co-written by A.R., M.K. and B.J.
