## [Peer Review File · Nature Communications]

Reviewers' Comments:

Reviewer #1 (Remarks to the Author)

The authors investigated the origin of fluorescence intermittency during the photoreduction of graphene oxide. They perform simulations to explain long time scale intermittency involved in these processes. The major concerns I have with the paper are primarily with the experimental data, the confusing discussion throughout the paper, and what appears to be an odd description of blinking. I would recommend publishing after major revisions and clarification of several key points. Specific comments are listed below.

The experimental conditions and even the sample measured are completely unclear in the manuscript. Are the authors measuring 20x20 nm graphene fragments like they model or are they measuring something else?

How do they perform their emission studies? Are these EMCCD based fluorescence measurements. If so why are there no images shown of the fluorescence? Or are they confocal based measurements scanning across the sample? Again why no images?

The authors state that figure 2C shows "close-up of emission profile during the blinking phase". It is not clear to me how this figure shows blinking in the experimental or the calculated trace. Blinking is an inherently single molecule process. The emission of this figure shows fluorescence gradually increasing over the course of 100s of seconds and then gradually decreasing over the same time scale. There are some features in the spectra but it is not clear they are outside of the range of poisson noise. I don't understand how this illustrates any blinking behavior. There is no description of the sample so it is not clear if they are representing that there are domains within the graphene that are exhibiting blinking. If that is the case this figure makes even less sense. There needs to be a much more thorough description of what they are measuring and how this data in any way constitutes blinking behavior.

They reference Lu et al (#5) as an example of blinking but they list it as published in 2010 and this was published in 1998. Also the paper did not measure single cholesterol molecules as the authors claim here. Lu et al measured single molecule behavior of a protein called cholesterol oxidase. The authors should look at figure 1b of Lu et al for an example of a single molecule emission time trace which is characterized by transitions between a fluorescent and non-fluorescent state.

Is the entire area of the graphene sheet fluorescent and there are regions that undergo intermittency? If so what is measured all of the fluorescence or spatially resolved fluorescence.

The modeling seems to be more clearly presented, but it would be helpful if they showed an emission trajectory of single domain that presumably would exhibit fluorescence intermittency? If they are modelling multiple domains simultaneously or modeling a single blinking domain and there is fluorescence that is generated from the sheet it is not clear how they would determine if there was blinking. It is not clear why all the isolated domains would be exactly in phase and exhibit identical blinking.

The convention that the authors use in sentences such as, "The emission (absorption) then exhibits a second photobrightening peak with blinking (progressive decay) prior to apparent" where they use parenthesis is confusing. I would advise dropping the parenthesis and write 2 separate or one conjugated sentence.

Reviewer #2 (Remarks to the Author)

The manuscript "Fluorescence intermittency..." authored by A.Ruth et al. relates to the very interesting phenomenon of fluorescence blinking in 2D-medium – single layer reduced graphene oxide (rGO). This effect was discovered a few years ago and already published (also with contribution of coauthors of the present manuscript) in a few papers (refs.4,11,12).

The main goal of the present paper is explaining of the microscopic nature of the above effect. For this authors consider the fluctuations of number/sizes emitting graphenic nanoclusters due to redistribution of carbon sp² domains. Authors performed theoretical modeling of GO-rGO conversion using DFT and Monte-Carlo techniques. Perfectly the results of calculations coincide with experimental data (emission/absorption trajectories, power spectral density).

The most impressive results are presented in supplementary materials in the form of video of reclustering effect together with animated graph of luminescence evolution.

In fact the effect of luminescence blinking in quantum confined media (2d-, 1d-, 0d-) is of extremely broad interest due to its fundamental meaning as well as potential applications in quantum optics, material science, biophysics. In some cases the effect should be suppressed (e.g. non-classic light sources), whereas in other application blinking is useful (labels for nanoscopy). Best, if it will be possible to control fluorescence intermittence. For this we need to understand the microscopic nature of the phenomenon. The authors of the reviewed manuscript performed very important step on this way. From this point of view a very broad scientific community is interested in reading the paper.

1. In my opinion the main shortcoming is that the main fundamental result was already somehow published in hi-ranking journal NanoLetters (ref.4 in 2015, ref.11 in 2014 and ref. 12 in 2013). In fact, the main experimental observation was reported there – blinking fluorescence in single layer reduced graphene oxide (rGO). The power spectral density was also founded. The heterogeneous nature of the observed phenomena was declared and attributed to the numerous switching channels. Actually, the main novelty of the presented manuscript is comprehensive explanation of the fluorescence intermittency as result of reclustering in rGO related to the redistribution of carbon sp² domains by using of computer simulation and methods of DFT.

2. Authors also state, that they present the first study with direct relation of blinking process to the definite physical (physical-chemical) process in quantum emitter. Actually it is not so univocal. For example, in the paper [Noah J. Orfield, ACS NANO, 10.1021/nn506420w (2015)] authors found correlation between defects in QD observed with electronic microscopy on the atomic scale with blinking parameters for concrete QD luminescence.

3. Finally, authors suggest some generalization of their conclusions for other fluorophores, namely for explanation of luminescence fluctuations over timescales much longer than typical fundamental electronic processes. They also state about light-induced (LI) structural changes as a common mechanism of blinking for emitters of different nature. In principle, LI mechanisms of blinking is well known, however the microscopic physical-chemical nature of the process can be different in the different kind of emitters (see e.g. Orlov et al., J. Chem.Phys. 137, 194903, 2012). This question should be discussed in the manuscript in more details with appropriate referencing.

4. One of the most common theoretical approach to explain blinking phenomena (especially in semiconductor nanocrystals) is charging model (Efros, Rosen, Phys. Rev. Lett. 78, 1110, 1997) and its modifications (e.g. Osad'ko et al, J. Phys. Chem. C, 119, 22646, 2015). I think that all the findings of the presented manuscript should be compared with these models also with corresponding references.

In conclusion, the manuscript Ruth et al. "Fluorescence intermittency..." considers the very important question which is interesting for very broad interdisciplinary scientific community. The topic is appropriate for Nature Communication, as well as level of performed work. The main disadvantage is that there are no new experimental discoveries in the paper. All the pioneer experimental results were published in previous papers in hi-ranking journals. Also there is some inaccuracy in description of relations with theories and experimental data obtained by other groups. The generalization of approach to other fluorophores seems to be not convincing. Based on the above statements, I would suggest the authors to revise their manuscript before a final decision is reached, or publish it in NanoLetters as logical continuation of Ref.4.

Reviewer #3 (Remarks to the Author)

The submitted paper deals with the very substantial and timely issue of luminescence blinking and photo bleaching/blueing during the photoreduction of graphene oxide. Experimental results are compared with elaborate calculations at different length and time scales making use of DFT and MC. The simulated reclusterings compare convincingly with experimental findings. These are very promising (and one of the very few) approaches to relate blinking to a mechanistic model via fluctuations of physico-chemical properties. The paper deserves without any doubt publication in Nature Communications. A few considerations should be taken into account.

- 1) The authors argue that their findings can be transferred to other quantum systems. However, the investigated system is a kind of irreversible reaction which is only partly the case for e. g. QDs. In consequence, fluctuating reactions should be taken also into account.
- 2) What happens if luminescent clusters have not only radiative but also non-radiative pathways?
- 3) The authors discuss the influence of reversible reactions with respect to spectral densities. Obviously they did not take them into account. Why not? This would make their work more close to a general model of blinking, also for other quantum systems.
- 4) It is not clear on which base authors ordered $k(C1)$ to $k(C3)$ in Equ.(3)

Minor Points:

- 5) Authors should give more general examples instead of Ref (1).
- 6) It seems that in the discussion of Fig. 2b the assignment of the plateau either to absorption and emission has been intermixed in the text.

Response to Reviewer's comments

We address each specific comment and suggestion (original reproduced here in italics), and answer each one in turn.

Reviewer #1 (Remarks to the Author):

The authors investigated the origin of fluorescence intermittency during the photoreduction of graphene oxide. They perform simulations to explain long time scale intermittency involved in these processes. The major concerns I have with the paper are primarily with the experimental data, the confusing discussion throughout the paper, and what appears to be an odd description of blinking. I would recommend publishing after major revisions and clarification of several key points. Specific comments are listed below.

1. The experimental conditions and even the sample measured are completely unclear in the manuscript. Are the authors measuring 20x20 nm graphene fragments like they model or are they measuring something else?

The relative size between the fluorescent domains that we model and the experimental sample is provided by the scale bars in figure 1. In particular, a typical DFT fluorescent domain is a few angstroms. Monte Carlo (MC) simulations, on the other hand, consider a 20x20 nm area. Finally, the experiment measures the spatially-resolved emission from a GO flake that is roughly 4x4 microns in size. Since these are far field measurements, the experimental spatial resolution is roughly half the wavelength of light. When the emission is captured with an electron multiplying CCD camera, the resulting (experimental) spatial resolution is approximately 200 nm/pixel. Thus, each pixel of an experimental movie represents a 200x200 nm area. Consequently, to simulate *one* experimental pixel, 100 MC simulations are averaged together. Note that details regarding the experiment have previously been provided in references 20 and 21. However, to make the current manuscript as self-contained as possible, a full description of experimental details has been added to the Methods section. Specifically, we now include the following new text:

“Synthesis of Graphene Oxide

GO was synthesized using a modified Hummers method.^[21] In brief, 300 mg of graphite, 36 ml of H₂SO₄, and 4 ml of H₃PO₄ were combined in an ice bath and continuously stirred for 4 hours. Next, 3.6 mg of KMnO₄ was added and the mixture was stirred for an additional 48 hours. The temperature was kept below 25 °C for this period. The suspension was then diluted to double the volume with deionized (DI) water and subjected to sonotrode sonication at 20 kHz for 1.5 hours. Finally, 3% H₂O₂ was added to the suspension until it turned yellow, indicating that highly oxidized graphite was produced. Upon centrifugation, the precipitate was washed with 1 M HCl and then repeatedly washed with DI water. The resulting red-brown suspension was then freeze-dried for storage, yielding a cream colored solid.^[21]

Individual GO Sheet Microscopy

Dilute GO/ethanol suspensions were sonicated at 42 kHz for ~10 s and subsequently drop-cast onto flamed fused silica microscope coverslips. This produced a sample coverage of ~1 single layer GO sheet per several μm^2 , with flake sizes ranging from 1 to 10 μm . Individual

sheets were imaged with a home-built inverted microscope utilizing a continuous wave 405 nm laser. The excitation source was introduced to the sample in an epi-illumination arrangement with an associated excitation intensity of 380 W cm^{-2} and an approximate $30 \text{ }\mu\text{m}$ excitation spot. Fluorescence was captured with an electron-multiplied charge-coupled device (EM-CCD) *via* a collection objective (Zeiss, 1.4 N.A.) and imaging lens ($f=160 \text{ mm}$) for a final resolution of approximately 200 nm/pixel .^[21]

In addition, we now include an introductory paragraph at the beginning of the Methods section that more intuitively explains the direct comparison we make between our developed model and previous experimental results. Specifically, on p. 9 we have included the following text:

“In order to provide molecular insight into optically measured experimental data, the developed model scales six orders of spatial magnitude in a series of steps. First, DFT and TDDFT is carried out to determine gap energies, radiative recombination rates, nonradiative recombination rates, and thus quantum yields on \AA - nm scaled fluorescent domains (cf. Fig. S1). The domains we studied via DFT/TDDFT span the entire visible range and are representative of all sp^2 domains which contribute to the photoluminescence of graphene oxide. Next, MC simulations use this information to model larger regions of a GO sheet during dynamical chemical reactions. During the simulation, cluster detection is used to count the number of connected carbon atoms; the gap energy, associated absorption cross-section, and quantum yield expressions are applied to calculate the color, absorption, and emission intensity of that cluster. Clusters whose gap energy exceeds the photon excitation energy are excluded. The absorption cross-section and emission intensity of all clusters within the region is summed up. The absorption is rescaled to calculate a percent absorption. Finally, in order to simulate a region comparable to a pixel in the experiment, many small regions are simulated and the results are averaged.”

2. How do they perform their emission studies? Are these EMCCD based fluorescence measurements. If so why are there no images shown of the fluorescence? Or are they confocal based measurements scanning across the sample? Again why no images?

In brief, the emission studies are performed in an epi-illumination (widefield) arrangement. This means that individual GO sheets are illuminated with light and the resulting emission is imaged with an EMCCD camera. By simply illuminating the sample with a 405 nm excitation source (380 W/cm^2), we find that GO's emission intensity photobleaches within the first ~ 30 seconds. Continued illumination (hundreds of seconds) results in fluorescence recovery and eventually grows up to 3x more intense than the initial emission intensity. A strong blinking behavior is observed during this brightening phase, wherein each spatially-resolved area ($200 \times 200 \text{ nm}$) seemingly blinks independently from the rest of the sheet. Next, the sheet exhibits a second photobleaching phase (starting at around 1000 seconds), in which fluorescence is irreversibly quenched, and the signal never recovers. All of these observations are detailed in references 20-22.

Movies of this dynamic process are acquired and are subsequently analyzed using ImageJ. Note that a fluorescence image from one such movie has been provided in figure 1 (d) of the main text. In response to the above reviewer comment, we have now included explicit experimental details to the Methods section of this manuscript as outlined in our response to comment 1.

3. The authors state that figure 2C shows “close-up of emission profile during the blinking phase”. It is not clear to me how this figure shows blinking in the experimental or the calculated trace. Blinking is an inherently single molecule process. The emission of this figure shows fluorescence gradually increasing over the course of 100s of seconds and then gradually decreasing over the same time scale. There are some features in the spectra but it is not clear they are outside of the range of poisson noise. I don’t understand how this illustrates any blinking behavior. There is no description of the sample so it is not clear if they are representing that there are domains within the graphene that are exhibiting blinking. If that is the case this figure makes even less sense. There needs to be a much more thorough description of what they are measuring and how this data in any way constitutes blinking behavior.

In response to this concern, we first ask the reviewer to view the attached video from our previous paper: DOI: 10.1021/acs.nanolett.5b00191. We have obtained permission to reuse this video and will now include it in the supplementary information of the current article. Next, beyond visual confirmation of blinking, we quantitatively establish its existence through an analysis of corresponding power spectral densities. Specifically, we find that experimental power spectral densities fit well to power law expressions of the form $PSD(\omega) = c \omega^{-\alpha}$. The power law exponent during the blinking phase of acquired movies is close to 1. This observation of a power law PSD is near universal in blinking studies. See, for example, references 4-8. Observation of power law PSDs therefore provides indirect evidence for blinking in GO.

Next, we explicitly reveal the blinking by Fourier filtering both the experimental emission trajectory in Figure 2c and its corresponding MC trajectory. The filtered data is shown below in the accompanying plot which is also Figure 3 of the revised manuscript. In particular, experimental and theoretical traces from Figure 2c are Fourier filtered to remove low (0.01 Hz cut-on) frequency components to exclude slowly varying photobleaching/photobrightening effects present in both the experiment and simulation.

In the resulting filtered data, both experimental (red line) and theoretical (blue line) trajectories show clear evidence of fluorescence intermittency beginning at ca. 500 s and ending at ca. 3000 s. Because the Monte Carlo simulation captures the essential behavior present in experimental emission and absorption trajectories, we conclude that the structural evolution occurring in the experiment is the same as that being simulated. Specifically, the Monte Carlo simulation indicates that GO undergoes a reclustered process whereby photolytic chemistries (as represented by Equations 1-3 of the main text) lead to the rapid generation and destruction of small emissive domains, resulting in apparent fluorescence intermittency. On page 1 of the SI, we now add:

“VideoExperimentalBlinking.mp4 shows blinking in grapheme-oxide captured by CCD camera.”

Additionally, VideoBlinking.mp4 was renamed VideoMonteCarloBlinking.mp4 to prevent confusion with the experimental video.

To address the reviewer’s concern, we now add the following text to page 6 of the manuscript.

“Figure 3 shows the experimental and MC trajectories after Fourier filtering. This process removes the long-term evolution of GO’s emission and more clearly reveals the fluorescence intermittency that is present. Blinking begins and ends at approximately the same time in the MC and the experiment. This leads us to conclude that the MC simulation captures the key difference between the stable emission and the fluorescence intermittency phase of the system.”

Furthermore, we added a new figure 3 and corresponding caption on page 7 of the main text

Figure 3. Onset of blinking revealed by Fourier filtering of experimental and Monte Carlo power spectral densities. In both cases, frequencies below 0.01 Hz have been removed. The envelope of the experimental Fourier filtered trajectory grows around ca. 500 s and decays around ca. 3000 s. The magnitude of the Monte Carlo Fourier filtered trajectory grows and decays at nearly the same time as the experiment.

4. They reference Lu et al (#5) as an example of blinking but they list it as published in 2010 and this was published in 1998. Also the paper did not measure single cholesterol molecules as the authors claim here. Lu et al measured single molecule behavior of a protein called cholesterol oxidase. The authors should look at figure 1b of Lu et al for an example of a single molecule emission time trace which is characterized by transitions between a fluorescent and non-fluorescent state.

We thank the reviewer for pointing out these errors. We have corrected the citation and have updated the text to reflect the molecular system actually being studied.

5. Is the entire area of the graphene sheet fluorescent and there are regions that undergo intermittency? If so what is measured all of the fluorescence or spatially resolved fluorescence.

The entire GO sheet is excited simultaneously with a laser in an epi-illumination configuration. The resulting fluorescence is imaged with an inverted microscope coupled to an EMCCD camera at 200 nm/pixel resolution. The entire GO sheet is fluorescent as seen from the fluorescence image in figure 1d. Intriguingly, we observe heterogeneous fluorescence throughout GO's basal plane—including a dynamic fluorescence response and heterogeneous blinking statistics (videos of these phenomena can be found in refs 20-22, and the Supplementary Movie provided with this manuscript. More specifically, there is a photobleaching/photobrightening effect within the whole sheet, and each section of the sheet seems to blink of its own accord. These initial observations led to three studies (as reviewer #2 points out, refs 20-22) that deal with the experimental results (refs 20 and 21) and blinking statistics (ref 22). The current manuscript ties all of these investigations together by providing a self consistent analysis that establishes causality between blinking and an actual chemical process which occurs on GO's basal plane during its photoreduction.

We have addressed much of this experimental concern in point 1 above and refer the reviewer to that discussion as well as the revised Methods section for more details.

6. The modeling seems to be more clearly presented, but it would be helpful if they showed an emission trajectory of single domain that presumably would exhibit fluorescence intermittency? If they are modelling multiple domains simultaneously or modeling a single blinking domain and there is fluorescence that is generated from the sheet it is not clear how they would determine if there was blinking. It is not clear why all the isolated domains would be exactly in phase and exhibit identical blinking.

There are two scenarios in which blinking in the 200x200 nm region becomes experimentally observable. In the first scenario, all domains contained in the region blink in phase. As the reviewer has also suggested above, this would be extremely unlikely. In the second scenario, the domains blink independently of each other and hence possess random phases. This also leads to blinking which becomes especially apparent when relatively few emissive domains exist in the 200x200 nm region. We are suggesting this latter scenario.

Note that individual blinking domains are below the optical diffraction limit and hence cannot be resolved experimentally. However, a small collection of blinking domains does lead to a trajectory like that seen in Figure 2c. By carrying out Fourier filtering of the collective trajectory

we reveal the underlying blinking which exists. This is illustrated in Figure 3 of the revised main text and is discussed in greater detail above in comment 3.

7. The convention that the authors use in sentences such as, “The emission (absorption) then exhibits a second photobrightening peak with blinking (progressive decay) prior to apparent ...” where they use parenthesis is confusing. I would advise dropping the parentheses and write 2 separate or one conjugated sentence.

We have removed this stylistic convention from the manuscript and have replaced each instance with two separate sentences. The edited text on page 5 is reproduced below:

“... the emission exhibits an initial decay, followed by a low plateau. Meanwhile the absorption exhibits an initial rise followed by a high plateau. The emission then exhibits a second photobrightening peak with blinking prior to apparent photobleaching and irreversible sample damage. While the emission is intermittent, the absorption follows a smooth decay.^[20]”

Reviewer #2 (Remarks to the Author):

The manuscript “Fluorescence intermittency...” authored by A.Ruth et al. relates to the very interesting phenomenon of fluorescence blinking in 2D-medium – single layer reduced graphene oxide (rGO). This effect was discovered a few years ago and already published (also with contribution of coauthors of the present manuscript) in a few papers (refs.4,11,12) {20-22}. The main goal of the present paper is explaining of the microscopic nature of the above effect. For this authors consider the fluctuations of number/sizes emitting graphenicnanoclusters due to redistribution of carbon sp² domains. Authors performed theoretical modeling of GO-rGO conversion using DFT and Monte-Carlo techniques. Perfectly the results of calculations coincide with experimental data (emission/absorption trajectories, power spectral density). The most impressive results are presented in supplementary materials in the form of video of reclustering effect together with animated graph of luminescence evolution. In fact the effect of luminescence blinking in quantum confined media (2d-, 1d-, 0d-) is of extremely broad interest due to its fundamental meaning as well as potential applications in quantum optics, material science, biophysics. In some cases the effect should be suppressed (e.g. non-classic light sources), whereas in other application blinking is useful (labels for nanoscopy). Best, if it will be possible to control fluorescence intermittency. For this we need to understand the microscopic nature of the phenomenon. The authors of the reviewed manuscript performed very important step on this way. From this point of view a very broad scientific community is interested in reading the paper.

1. In my opinion the main shortcoming is that the main fundamental result was already somehow published in hi-ranking journal NanoLetters (ref.4 {22} in 2015 , ref.11 {20} in 2014 and ref. 12 {21} in 2013). In fact, the main experimental observation was reported there – blinking fluorescence in single layer reduced graphene oxide (rGO). The power spectral density was also founded. The heterogeneous nature of the observed phenomena was declared and attributed to the numerous switching channels. Actually, the main novelty of the presented manuscript is comprehensive explanation of the fluorescence intermittency as result of reclustering in rGO

related to the redistribution of carbon sp² domains by using of computer simulation and methods of DFT.

We would like to thank Reviewer #2 for pointing out the importance of our theoretical results and for agreeing with us that our results deserve the attention of the broad audience served by Nature Communications. While the experimental results have previously been published and a phenomenological MRC model has been applied to analyze blinking in this system, the microscopic mechanism for blinking was never established. Note that the MRC model simply states that there are recombination centers responsible for blinking, which are distributed uniformly in time on a logarithmic scale. The phenomenological MRC model does not specify what these recombination centers are. Thus, the main novelty and importance of the current study is that - in contrast to the phenomenological MRC framework we have adopted previously - we *directly* establish the causality between blinking and actual GO photolytic chemistries in a microscopic, semiquantitative manner.

To address the Reviewer's concern, we have altered the following text from page 2

“Intriguingly, the brightening phase is punctuated by spatially-heterogeneous fluctuations in emission intensity with 1/f power spectral densities (PSDs) typical of fluorescence intermittency.^[3]”

The new text is:

“Intriguingly, the brightening phase is punctuated by spatially heterogeneous fluctuations in emission intensity with 1/f power spectral densities (PSDs) typical of fluorescence intermittency.^[22] Although the multiple recombination center model^[23] provides a phenomenological framework for the 1/f power spectrum observed experimentally, it does not provide a microscopic mechanism for blinking.”

Reference added:

[23] Frantsuzov, P. A., Volkán-Kacsó, S. and Jankó, B. Model of Fluorescence Intermittency of Single Colloidal Semiconductor Quantum Dots Using Multiple Recombination Centers. *Phys. Rev. Lett.* **103**, 207402 (2009).

2. Authors also state, that they present the first study with direct relation of blinking process to the definite physical (physical-chemical) process in quantum emitter. Actually it is not so univocal. For example, in the paper [Noah J. Orfield, ACS NANO, 10.1021/nn506420w (2015)] authors found correlation between defects in QD observed with electronic microscopy on the atomic scale with blinking parameters for concrete QD luminescence.

We thank the reviewer for drawing our attention to the very interesting recent study by Orfield and his collaborators. They correlate the statistical average of the on-time (normalized by the length of the trajectory) of an individual quantum dot with its TEM image. Then, by segmenting 84 total quantum dots into “high” and “low” emitters, Orfield establishes structural reasons for why some quantum dots exhibit a large fractional on-time and others a low fractional on-time. In particular, Orfield's statistical/structural study finds that:

1. If the quantum dot core is unpassivated with an inorganic shell, the on fraction is low or zero;
2. If there is a zincblende stacking fault at the core/shell interface, on fractions are low;
3. If there is a Cd face on the core that is not well passivated, then on fractions are low;
4. If the core is generally well passivated by the shell all around, on fractions are large.

Although Orfield et al., observe correlations between structure and fractional on/off times, they do not: (a) provide a microscopic theoretical model for the defect, (b) show how the defect leads to blinking, (c) generate a theoretical emission trajectory, and (d) show how the power spectral density of their theoretical trajectory exhibits 1/f-like behavior. Any viable microscopic model of fluorescence intermittency must be able to provide all the results (a)-(d). In contrast to Orfield's paper, and - as far as we know - any other studies currently available, our study provides all the necessary ingredients mentioned above.

Finally, the Orfield manuscript has one more concerning issue. Given that on-time and off-times are power law distributed and have different power law coefficients, it stands to reason that the fractional on-time measured by Orfield et al. varies depending on what timescale one is looking at. To illustrate, if on-times have a power law slope of $m_{on} = 1.9$ and off-times have a power law slope of $m_{off} = 1.7$, then at a longer experimental integration time one will see a large fractional off-time and correspondingly a small fractional on-time. With a shorter experimental integration time one sees the reverse. Namely, the fractional on-time becomes larger than the fractional off-time. So fractional on/off-times are not really meaningful parameters in a deeper sense given the existence of power law kinetics. Furthermore, in our 2009 PRL, we demonstrated that the on- and off-time power law exponents depend strongly on the choice of the threshold - this is precisely the reason why we decided to study the threshold-independent power spectral density as the indicator of blinking in nanoscale fluorophores.

To address the Reviewer's comment we have made the following changes to the manuscript. Specifically, the following two statements have been modified as follows.

Original:

1. (In the abstract) "These simulations thus provide the first direct link between currently unexplained, long timescale emission intermittency in a quantum mechanical fluorophore and a physical process leading to its structural/chemical change."
2. (Page 2) "...the current study is the first study to link blinking to a definable physical and/or chemical process."

New:

1. (In the abstract) "These simulations thus establish causality between currently unexplained, long timescale emission intermittency in a quantum mechanical fluorophore and identifiable chemical reactions that ultimately lead to switching between on and off states."

2. (Page 2) "... the current study is the first study to link blinking to a definitive chemical process."

3. *Finally, authors suggest some generalization of their conclusions for other fluorophores, namely for explanation of luminescence fluctuations over timescales much longer than typical fundamental electronic processes. They also state about light-induced (LI) structural changes as a common mechanism of blinking for emitters of different nature. In principle, LI mechanisms of blinking is well known, however the microscopic physical-chemical nature of the process can be different in the different kind of emitters (see e.g. Orlov et al., J. Chem.Phys. 137, 194903, 2012). This question should be discussed in the manuscript in more details with appropriate referencing.*

We would like to thank Reviewer #2 for pointing out the very interesting study performed by Orlov and coworkers. They document the blinking process in a single dye molecule (tetra-*tert*-butylterrylene) in an amorphous matrix (polisobutylene) at cryogenic temperatures. Indeed, as the reviewer points out, light-induced structural changes have been seen before, but surprisingly, temperature plays an important role in the system studied by Orlov et al. This in contrast to many other nanoscale blinkers, such as self-assembled quantum dots, colloidal quantum dots, rods, wires, single molecules in a matrix, etc.; (see for example, our Ref. [4] in the current version of the manuscript, which presents an overview of seven different classes of emitters). Typically, the energy scales governing the blinking process (including the energy of the incoming photons) are much larger than the ambient temperature. Orlov's paper is a good illustration for why we do not think there a universal mechanism for blinking, and we agree that the details of the blinking process can be different for different systems. However, we think there must be common underlying features across the various classes of blinkers, especially if we compare their universal features, such as the emergence of intensity fluctuations over times scales that are much longer than typical electronic time scales of fluorophores. In this manuscript we provide an example of a microscopic mechanism for blinking, in which a slowly varying classical quantity, directly controlling the emission intensity, emerges from the description of an essentially quantum mechanical system. In the present case, the quantity is the number of optically active clusters. We believe that it is this feature of our mechanism, the emergence of a new quantity which controls emission, which leads to a power-law frequency spectrum.

In order to clarify this point, we have modified the main text of the manuscript. Specifically, we now include the paper of Orlov et al in our list of references, and on page 8 the text was changed from

"Such light-induced structural changes are also the likely cause of blinking in other fluorophores."

to

"Such light-induced structural changes are also the likely cause of blinking in other fluorophores^[35] although there may not be a universal mechanism for blinking."

Furthermore, we would like to point out to Reviewer #2 the new sections in the manuscript that were included following the suggestion by Reviewer #3, in which we discuss the role of reversible chemical processes, and their potential for generalizing the current strategy of building a microscopic model for other nanoscale emitters.

4. One of the most common theoretical approach to explain blinking phenomena (especially in semiconductor nanocrystals) is charging model (Efros, Rosen, Phys. Rev. Lett. 78, 1110, 1997) and its modifications (e.g. Osad'ko et al, J. Phys. Chem. C, 119, 22646, 2015) I think that all the findings of the presented manuscript should be compared with these models also with corresponding references.

In conclusion, the manuscript Ruth et al. "Fluorescence intermittency..." considers the very important question which is interesting for very broad interdisciplinary scientific community. The topic is appropriate for Nature Communication, as well as level of performed work. The main disadvantage is that there are no new experimental discoveries in the paper. All the pioneer experimental results were published in previous papers in hi-ranking journals. Also there is some inaccuracy in description of relations with theories and experimental data obtained by other groups. The generalization of approach to other fluorophores seems to be not convincing.

Based on the above statements, I would suggest the authors to revise their manuscript before a final decision is reached, or publish it in NanoLetters as logical continuation of Ref.4 {22}.

As Reviewer #2 correctly pointed out above, our work is indeed a proposal for a microscopic mechanism of blinking. As we mentioned before in our response to comment 1 by Reviewer #2 the strategy and the methods we employed in this manuscript is in sharp contrast to those we used in our previous papers, based on the phenomenological MRC model. Here we started out from our density functional theory (DFT) results for microscopic graphene clusters; second, we used the DFT results to build a Monte Carlo simulation for the evolution of clusters in all phases of the reduction process; third, we used the simulations to calculate theoretical trajectories; finally, we showed that the theoretical trajectories (emission/absorption correlations, power spectral densities, etc.) are in semi-quantitative agreement with their experimental counterparts. As such, this microscopic approach is a crucial step towards understanding and eventually controlling blinking. Indeed, as Reviewer#2 wrote "The authors of the reviewed manuscript performed very important step on this way. From this point of view a very broad scientific community is interested in reading the paper." We therefore strongly believe that our theoretical results and proposal for a microscopic mechanism of blinking deserves the broad audience served by Nature Communications.

In response to the concern expressed in this comment by Reviewer #2, we have now added an additional background section to the manuscript's introduction which discusses the charging model. Specifically, we add on page 2 the following text.

"Many models have been developed to explain the contrast between fast radiative recombination (ns) and slow timescales seen in fluorescence intermittency (s-min).^{[4][5][6][7][8]} One of first and most widely used microscopic models assumes that the dark state is charged^{[9][10]}, and it is often invoked to justify the existence of long intermittent periods in colloidal quantum dot blinking

trajectories. In this model, an on-to-off transition occurs due to Auger ionization of a nanocrystal. The implicit assumption is that a charged quantum dot remains non-emissive until neutralized by return of the ejected electron. Although some of the predictions of the original Efros-Rosen model are in contradiction with experiments^{[11][12]}, recent work by Osad'ko et al^[10] showed that a modified version of the charging model is consistent with power law kinetics. However, as shown by Guyot-Sionnest^[11], charged quantum dots are emissive, and alternative blinking scenarios also have to be considered. Quantum dot blinking has also been attributed to light-induced surface ligand reorganization^[13] and even to light-induced defect rearrangement in nanocrystals.^[14] Beyond quantum dots, blinking has been observed in a host of other systems. Examples includes instances of fluorescence intermittency in single molecules^[15] and in single polymer strands.^[16] Explanations for blinking in these other systems hence range from surface electron (hole) traps to oxidation. Consequently, within the broader context of single particle microscopy, and even within the realm of quantum dot blinking, there exists no consensus as to the microscopic origin of blinking. Our paper provides a definitive link between blinking in rGO and photostimulated carbon etching.”

In this discussion, the following new references were added:

- [9] Efros, A. L., Rosen, M. Random Telegraph Signal in the Photoluminescence Intensity of a Single Quantum Dot. *Phys. Rev. Lett.***78**, 1110-1113 (1997).
- [10] Osad'ko, I. S., Distribution of Photons in Single Quantum Dot Intermittent Photoluminescence with Power-Law Distribution of On/Off Intervals. *J. Phys. Chem. C.*, **117**, 11328-11336 (2013).
- [11] Jha, P. P. &Guyot-Sionnest, P. Trion Decay in Colloidal Quantum Dots. *ACS Nano***3**, 1011–1015 (2009).
- [12] Zhao, J., Nair, G., Fisher, B. R., &Bawendi, M. G. Challenge to the Charging Model of Semiconductor-Nanocrystal Fluorescence Intermittency from Off-State Quantum Yields and Multiexciton Blinking. *Phys. Rev. Lett.***104**, 157403 (2010)
- [13] Voznyy, O., Thon, S. M., Ip, A. H., & Sargent, E. H. Dynamic Trap Formation and Elimination in Colloidal Quantum Dots *J. Phys. Chem. Lett.***4**, 987–992 (2013).
- [14] Voznyy, O. & Sargent, E. H. Atomistic Model of Fluorescence Intermittency of Colloidal Quantum Dots. *Phys. Rev. Lett.***112**, 157401 (2014).
- [15] Mason, M. D., Credo, G. M., Weston, K. D. &Burrato, S. K. Luminescence of Individual Porous Si Chromophores*Phys. Rev. Lett.***80**, 5405 (1998).
- [16] Vanden Bout, D., Yip, W. T., Hu, D., et al. Discrete Intensity Jumps and Intramolecular Electronic Energy Transfer in the Spectroscopy of Single Conjugated Polymer Molecules. *Science***277**, 1074 (1997).

Reviewer #3 (Remarks to the Author):

The submitted paper deals with the very substantial and timely issue of luminescence blinking and photo bleaching/blueing during the photoreduction of graphene oxide. Experimental results are compared with elaborate calculations at different length and time scales making use of DFT and MC. The simulated reclustering compares convincingly with experimental findings. These are very promising (and one of the very few) approaches to relate blinking to a mechanistic model via fluctuations of physico-chemical properties. The paper deserves without any doubt publication in Nature Communications. A few considerations should be taken into account.

We thank Reviewer #3 for his/her insightful comments and for agreeing with us that fluorescence intermittency in rGO deserves the attention of the diverse readership of Nature Communication.

1) The authors argue that their findings can be transferred to other quantum systems. However, the investigated system is a kind of irreversible reaction which is only partly the case for e. g. QDs. In consequence, fluctuating reactions should be taken also into account.

We agree with Reviewer #3 that fluctuating reactions should be taken into account for QDs. Consequently, to address the Reviewer's comment, we have first modified our statement that the findings can be directly transferred to other systems. It has been changed from

(page 6)

“Experimental and theoretical power law exponents of $\alpha=0.66$ and $\alpha=1.89$ are found, which fall within the range of exponents (i.e. 0.5 to 2.0) typically seen for other nanoscale emitters exhibiting fluorescence intermittency.^[1] The quantitative discrepancy between experiment and theory is readily explained by the absence of reversible reactions faster than the lattice disintegration reactions considered in equations (1-3). These processes would add additional spectral weight to higher frequencies and, in turn, lead to smaller power law exponents (cf. SI).”

to

“Experimental and theoretical power law exponents of $0.5 < \alpha < 0.9$ and $\alpha = 1.7$ are found, which fall within the range of exponents (i.e. 0.5 to 2.0) typically seen for other nanoscale emitters exhibiting fluorescence intermittency.^[4] The quantitative discrepancy between experiment and theory is narrowed by including Poisson noise. In this regard, Poisson noise accounts for counting statistics and also provides a simple model for intensity fluctuations arising from reversible reactions. The latter, in particular, stems from dynamic changes to the number of emissive domains, which contribute to the overall observed emission intensity. Including counting statistics to the Monte Carlo simulation reduces its power law exponent to $\alpha = 0.7$, which is closer to the range of exponents seen in the experiment. Furthermore, after including cluster fluctuations to model reversible processes, we recover substantial spectral weight at high frequencies. Additional details regarding the effect of Poisson noise on the Monte Carlo results can be found in the SI.”

Next, following the Reviewer's comments and suggestion, we have added a new section to the SI where we consider reversible reactions in the modeling through an approximation (i.e. the

introduction of Poisson noise) and show what results to the power spectrum. In particular, we add in the SI the following new text.

“Section 4: Monte Carlo power spectrum with counting statistics and reversible processes

In order to account for both experimental counting statistics (shot noise) and the likely existence of reversible reactions, we consider the impact of introducing Poisson noise to the Monte Carlo results. In the former case, Poisson noise just models experimental shot noise. In the latter case, Poisson noise can be used to model the effects of chemical reversibility in equations (1-3) of the main text. Specifically, when reverse reaction timescales are comparable to the experimental integration time, variations in the number of active (emissive) domains will cause fluctuations to the overall integrated emission intensity. This will then manifest itself as apparent noise in the data. Hence, the inclusion of additional stochastic noise intensity variations to the Monte Carlo simulation approximates the tangible effects of chemical reversibility.

The following are implicit assumptions and limitations of the approximation that should be noted: (a) First, if reversible reaction rates are significantly faster than the experimental binning time (second timescale), intensity fluctuations stemming from these reversible reactions will be averaged out. (b) The stochastic noise approximation is based on many emitters with equal photoluminescence intensity, and it assumes that each switching event is a single emitter event. Therefore, the intensity distribution can deviate from a Poisson distribution. (c) Next, if reversible reaction rates are significantly slower than the experimental binning time, intensity fluctuations will exhibit time-correlations, which are not captured by Poisson noise. (d) The use of Poisson noise to represent the tangible effects of reversible reactions therefore implicitly assumes that reverse reaction rates exhibit timescales comparable to the experimental binning time. In the absence of detailed activation energies and a full accounting of *all* possible reversible reactions, beyond those associated with equations (1-3), an accurate estimation of relevant reverse reaction rates cannot be established. Thus, at present, a fully rigorous accounting of reversibility is beyond the scope of this study.

In practice, to include counting statistics, for each data point in the Monte Carlo trajectory, the number of photons that would have been counted in a detector is randomly generated by choosing a value from a Poisson distribution. The mean of the Poisson distribution is set to the number of counts (max ~3000) seen in the original MC trajectory. To include stochastic noise, associated with reversible reactions, the same methodology is applied. The number of active clusters is generated randomly by choosing a value from a Poisson distribution. The mean of the Poisson distribution in this case is given by the number of Monte Carlo-estimated emissive clusters (max ~300). Then the number of clusters is multiplied by the average counts/cluster to obtain a new estimate for the total number of counts observed. Fluctuations in cluster number thus more significantly impacts the Monte Carlo trajectories than variations due to counting statistics.

Figure S5 shows the impact of adding Poisson noise to Monte Carlo trajectories. Namely, adding counting statistics to the Monte Carlo trajectory reduces its power law exponent to $\alpha = 0.7$. When only noise from reversible processes is considered, the power law exponent becomes even smaller: $\alpha = 0.3$. It is therefore evident that improved modeling of reversible reactions will lead to better agreement between experiment and theory.

Figure S5. Monte Carlo power spectrum altered by introducing Poisson noise to account for counting statistics and reversible processes. The figure compares power spectral densities of the experiment, the MC simulation, the MC with counting statistics, and the MC with stochastic noise. In each case, power law exponents are shown. For the raw MC PSD, a line has been included to indicate the existence of data points below the shown y axis limits.”

2) *What happens if luminescent clusters have not only radiative but also non-radiative pathways?*

We have already considered non-radiative pathways in the calculation of quantum yields (QYs). This is shown in Table 1 of the SI which lists both radiative recombination rates and non-radiative recombination rates calculated for individual domains. The quantum yield is calculated by taking the ratio of the radiative to the total (i.e. radiative + non-radiative) decay rate. Next, since all radiative and non-radiative decay rates (nanoseconds) are significantly faster than the time between measurements (seconds) in the experiment, they are implicitly included in the MC simulation via employed QY values.

In order to avoid any other misunderstandings we have clarified this in the methods section on page 9:

“Quantum yields are calculated by comparing the rate of radiative recombination with the total recovery rate to the ground state.”

3) *The authors discuss the influence of reversible reactions with respect to spectral densities. Obviously they did not take them into account. Why not? This would make their work more close to a general model of blinking, also for other quantum systems.*

We agree with the reviewer that this would make the model being developed more general. However, it would also mean introducing additional parameters (i.e. reverse reaction rates) into the model, making our simulations less tractable. This would make the model less clear to the reader, and would leave our conclusions considerably more parameter-dependent. In view of the good qualitative agreement between the current model and the experiment, including (a) the

well-substantiated proof of reclustering as an extension of photobrightening/photobleaching, (b) the appearance of a power law PSD, and (c) the appearance of long timescale blinking events from reclustering, we defer a full and rigorous treatment of chemical reversibility in our simulations to a future study.

However, to address the reviewer's concern, we have done the following:

- (1) In the main text, we have discussed how Poisson noise can be used to represent the effects of chemical reversibility in our MC simulations, but we highlight the results obtained without this assumption. This is discussed in comment 1 above.
- (2) Next, in the SI, we have added a new section where we elaborate on how the reversibility of Equations 1-3 in the main text can be estimated by adding Poisson noise to our Monte Carlo trajectories. We then illustrate how this affects MC PSDs, causing corresponding power law exponents to converge on experimental power law exponents. Again, this is discussed in greater detail in our response to comment 1 above.

4) *It is not clear on which base authors ordered $k(C1)$ to $k(C3)$ in Equ.(3)*

In figures 2a and 2b and VideoReduction.mp4, an effective carbon removal rate constant, k_C , has been used. The resulting three-parameter model for photoreduction (i.e. k_{OH} , k_O , and $k_C = k_{C1} = k_{arm} = k_{zig} = k_{C3}$) allows us to first establish and connect features of experimental emission/absorption trajectories with underlying chemical reactions responsible for changes to GO's optical response (figure S4). We find that this simulation reproduces the observed photobrightening and photobleaching as well as evolution of GO's absorption, resulting from photoreduction.

Next, because the three parameter model establishes that rGO blinking arises *at the same time* as carbon removal it implies causality between carbon extracting chemistries and fluorescence intermittency. Consequently, in what follows, we consider carbon removal processes in greater detail (figures 2c, 2d, 3, 4, and VideoMonteCarloBlinking.mp4). Specifically, we expand the modeling accuracy by now distinguishing carbon removal rate constants based upon local structure. Carbon removal rate constants are therefore ordered using computed reaction barrier energies. We find that the reaction barrier is higher for a carbon with three neighboring atoms (15 eV) than for a corresponding carbon atom with two (zigzag or armchair) wherein the reaction barrier for a carbon atom at a zigzag edge (7.5 eV) is larger than that at an armchair edge (4.5 eV). Finally, a carbon with only 1 neighboring carbon atom (i.e. attached to the edge of a plane) possess a range of barrier energies spanning (0.5-2.5 eV). Consequently, the resulting model parameterization becomes: $k_{C1} > k_{arm} > k_{zig} > k_{C3}$ which replaces the previous parameter, k_C , in successive calculations.

To clarify this model parameterization to the reader and to prevent any future misunderstandings, the following text has been added to page 4 of the main text:

“Additional details regarding the parameterization can be found in the SI”

Additionally, the text on page 6 was modified:

“Given that blinking occurs during carbon sublimation, we focus on more realistically modeling this phenomenon by allowing carbon sublimation rate constants to differ. This accounts for the varied local structure found in GO/rGO [e.g. singly-coordinate sites versus armchair edges, zigzag edges, and triply-coordinated sites]. Specifically, we set $k_{C1} > k_{Carm} > k_{Czig} > k_{C3}$ in equation (3).”

The above text was changed to:

“Given that blinking occurs during carbon sublimation, we focus on more realistically modeling this phenomenon by allowing carbon sublimation rate constants to differ. This accounts for the varied local structure found in GO/rGO [e.g. singly-coordinate sites versus armchair edges, zigzag edges, and triply-coordinated sites]. Specifically, we set $k_{C1} > k_{Carm} > k_{Czig} > k_{C3}$ in equation (3), and this parameterization is used in simulations yielding figures 2c, 2d, 3, and 4.”

Minor Points:

5) *Authors should give more general examples instead of Ref (1).*

Additional citations were added to the abstract which contain more general examples than the previous ref (1) {This reference was removed}. The new references are:

[5] Stefani, F. D., Hoogenboom, J. P., Barkai, E., Beyond quantum jumps: Blinking nanoscale light emitters. *Phys. Today***62**, 34 (2009)

[6] Cichos, F., von Borczyskowski, C., Orrit, M., Power-law intermittency of single emitters. *Curr. Op. Coll. Int. Sci.***12**, 272-284 (2007)

[8] Efros, A. & Nesbitt, D. J. Origin and control of blinking in quantum dots. *Nature Nanotechnology* **11**, 661–671 (2016)

6) *It seems that in the discussion of Fig. 2b the assignment of the plateau either to absorption and emission has been intermixed in the text.*

We thank the reviewer for catching a mistake in the sentence:

“Evident in either case is that the emission (absorption) exhibits an initial decay (rise), followed by a high (low) plateau.”

We have disambiguated this sentence following the suggestion of reviewer 1. The text now reads:

“... the emission exhibits an initial decay, followed by a low plateau. Meanwhile the absorption exhibits an initial rise followed by a high plateau. The emission then exhibits a second photobrightening peak with blinking prior to apparent photobleaching and irreversible sample damage. While the emission is intermittent, the absorption follows a smooth decay.”

REVIEWERS' COMMENTS:

Reviewer #1 (Remarks to the Author):

The authors have adequately addressed my concerns with the manuscript. The inclusion of the video was helpful. There does seem to be a minor formatting issue where spaces have been removed throughout the document. For example line 171 and line 208 a space is removed between 2 words.

Reviewer #2 (Remarks to the Author):

I consider all the author's answers to all three reviewers. The changes which were done by authors clarify most of my questions. The manuscript in the present form explain very well the main results of the work, as well as overview the current knowledge about blinking of nanosized emitters. As I asked, the authors figured out the advantages and features of their results in comparison with other theoretical treatments of the quantum emitter blinking processes.

They performed now the comprehensive analysis of the current state of the topic and include the appropriate referencing. At this point I just recommend to change (or add to) the new ref.10 [Osad'ko, 2013] by later paper [Osad'ko et al., J. Phys. Chem. C, 2015, 119 (39), pp 22646–22652] which demonstrate not only theoretical description but also experimental confirmation of the model describing QD blinking.

In conclusion, I think that the paper in the present form is appropriate for the Nature Communications. Of course the paper does not present the new experimental results, however the modeling (especially presented in supplementary materials) is very impressive. Moreover I see, that Nat. Comm. published regularly the purely theoretical papers.

Thus I agree that the paper of Ruth et al. "Fluorescence intermittency..." can be published in Nat. Comm. with above mentioned change.

Reviewer #3 (Remarks to the Author):

The authors have from my point of view considerably improved the ms. It should be published as it is.

Reviewer #1 (Remarks to the Author):

The authors have adequately addressed my concerns with the manuscript. The inclusion of the video was helpful. There does seem to be a minor formatting issue where spaces have been removed throughout the document. For example line 171 and line 208 a space is removed between 2 words.

We thank the Reviewer for recommending the current version of the paper for publication, and for the time and effort the Reviewer invested throughout the review process. We followed the Reviewer's and the Editor's recommendation, and fixed several formatting problems in the manuscript.

Reviewer #2 (Remarks to the Author):

I consider all the author's answers to all three reviewers. The changes which were done by authors clarify most of my questions. The manuscript in the present form explain very well the main results of the work, as well as overview the current knowledge about blinking of nanosized emitters. As I asked, the authors figured out the advantages and features of their results in comparison with other theoretical treatments of the quantum emitter blinking processes.

They performed now the comprehensive analysis of the current state of the topic and include the appropriate referencing. At this point I just recommend to change (or add to) the new ref.10 [Osad'ko, 2013] by later paper [Osad'ko et al., J. Phys. Chem. C, 2015, 119 (39), pp 22646–22652] which demonstrate not only theoretical description but also experimental confirmation of the model describing QD blinking.

In conclusion, I think that the paper in the present form is appropriate for the Nature Communications. Of course the paper does not present the new experimental results, however the modeling (especially presented in supplementary materials) is very impressive. Moreover I see, that Nat. Comm. published regularly the purely theoretical papers.

Thus I agree that the paper of Ruth et al. "Fluorescence intermittency..." can be published in Nat. Comm. with above mentioned change.

The reference (now reference 8) has been changed as suggested by the Reviewer. We would like to thank the Reviewer for the useful suggestions, and for his/her time and effort throughout the reviewing process.

Reviewer #3 (Remarks to the Author):

The authors have from my point of view considerably improved the ms.It should be publishes as it is.

We thank the Reviewer for recommending our paper for publication. We would also like to thank him/her the time and effort he/she invested throughout the review process.